# BENCHMARKING MACHINE LEARNING METHODS FOR STOCK PREDICTION

## ABSTRACT

Machine learning has been widely applied to stock movement prediction. However, research in this field is often hindered by the lack of high-quality benchmark datasets and comprehensive evaluation methods. To address these challenges, we introduce *BenchStock*, a benchmark that includes standardized datasets from the two largest stock markets (the U.S. and China) along with an evaluation method designed to facilitate a thorough examination of machine learning stock prediction methods. This benchmark covers a range of models, from traditional machine learning techniques to the latest deep learning approaches. Using BenchStock, we conducted large-scale experiments predicting individual stock returns over three decades in both markets to assess both short-term and long-term performance. To evaluate the impact of these predictions in actual market conditions, we constructed a portfolio based on the predictions and used a backtesting program to simulate its performance. The experiments revealed several key findings that have not been reported: 1) Most methods outperformed the S&P 500 in the U.S. market but experienced significant losses in the Chinese market. 2) Prediction accuracy of a method was not correlated with its portfolio return. 3) Advanced deep learning methods did not outperform traditional approaches. 4) The performance of the models was highly dependent on the testing period. These findings highlight the complexity of stock prediction and call for more in-depth machine learning research in this field.

## 1 INTRODUCTION

Among various time-series forecasting tasks, stock return prediction is a trendy research topic due to its substantial real-world implications. With the success of machine learning in domains such as natural language processing (NLP) and computer vision (CV), researchers have become increasingly interested in applying these techniques to stock prediction. Methods based on most popular machine learning networks, including Recurrent Neural networks (RNNs) (Deng et al., 2009; Du et al., 2021), Graph Neural Networks (GNNs) (Sawhney et al., 2021) and Transformers (Wang et al., 2022), have been used to improve the accuracy on stock return prediction. While all these methods are claimed to achieve good results, it is hard to identify real progress from them due to following challenges.

The first challenge arises from diverse datasets used in different studies, making it difficult to compare results between them. In contrast to computer vision field, in which standardized benchmarks such as ImageNet (Deng et al., 2009) and CIFAR-10 (Krizhevsky & Hinton, 2009) provide a common ground for method evaluation, various market indices from different countries were tested in stock prediction research and no identical dataset was used for comparison. For instance, DA-RNN (Deng et al., 2009) used NASDAQ 100 from the U.S. market, while FactorVAE (Duan et al., 2022) used China's A-share market. This lack of benchmark datasets makes it challenging to directly compare the effectiveness of different methods.

The second challenge arises from the lack of data standardization in current research. Many studies lack a standardized approach for preparing the data. While model structures and experimental results are often discussed in detail, data preprocessing is usually mentioned only briefly. Given the notoriously low signal-to-noise ratio in financial data, this lack of transparency further undermines the reliability of the research. Data preprocessing for stock data is particularly complex due to events like stock splits, which can significantly impact outcomes. Additionally, researchers with

deep learning expertise but limited financial domain knowledge often rely on free sources for stock data. For instance, several studies (Sawhney et al., 2021; Wang et al., 2022; Xia et al., 2024) used U.S. market datasets derived from Google Finance. These datasets not only fail to account for key financial events like stock splits and dividends but also lack the adjustment factors necessary for proper handling. By examining the Google Finance datasets used in these studies, we discovered instances of abnormal price changes exceeding 100% in a single day due to stock splits. Such anomalies can create returns and losses that do not exist in reality and significantly affect the result. Consequently, the absence of a rigorous standardization process raises serious concerns about the validity and reliability of findings in previous studies.

The third challenge comes from the absence of a unified evaluation method. The diverse datasets used in existing studies often cover different short periods, leading to inconsistent and potentially misleading results. AdaRNN (Du et al., 2021) analyzed data from 2017 to 2019, while DA-RNN (Deng et al., 2009) was solely tested on 2016. Given the variability in model performance across different periods, short-term analysis can skew the understanding of a model's effectiveness in stock prediction. This problem is further aggravated by using different evaluation metrics. While some works used error-based metrics such as MSE (Deng et al., 2009), others employed metrics like information coefficient (IC), information ratio of IC (ICIR), RankIC, and RankICIR (Du et al., 2021; Duan et al., 2022). Although these metrics are related to prediction accuracy, they do not directly reflect a model's ability in generating returns. STHAN-SR, ALSP-TF and CI-STHPAN (Sawhney et al., 2021; Wang et al., 2022; Xia et al., 2024) are among the few works that evaluated their methods using realized returns and Sharpe ratios of the formed portfolios. However, these studies calculated returns by averaging the returns of the top five stocks without accounting for real-world trading factors such as transaction fees. Implementing such a strategy in a market like China, where the trading cost is 0.4% per trade (0.15% for buy orders and 0.25% for sell orders), implies that a portfolio's value could decrease by more than 60% in a year due to transaction fees. Consequently, the portfolio must constantly generate annual returns exceeding 60% to avoid incurring a loss, which is an almost impossible target in any stock market. Thus, a systematic and consistent evaluation method is essential to accurately and pratically gauge the strengths and weaknesses of these methods.

In this paper, we introduce a benchmark *BenchStock* for the stock prediction task in the machine learning domain, aiming to provide a convenient tool for future research. We systematically evaluated 23 existing methods, including models from traditional machine learning to the most recent advancement in deep learning, using the same datasets and a comprehensive evaluation framework. We created two datasets from reputable sources, CRSP (Center for Research in Securities) and CSMAR (China Stock Market and Accounting Research Database), representing price and volume features of the U.S. and the Chinese stock markets, respectively, with thorough preprocessing. These methods were evaluated extensively by simulating real-world trading scenarios within a backtesting program from Microsoft's Qlib (Yang et al., 2020) over three decades. The performances of these models were quantified with prevalently used metrics from finance industry, including annual return (AR), Sharpe ratio (SR), information ratio (IR) and maximum drawdown (MDD). This study ensures a robust and realistic assessment of each model's efficacy in stock prediction.

The experiments revealed several noteworthy findings that had not been previously reported. First, while most methods performed well in the U.S. market, this success did not extend to the Chinese market. Although nearly all methods outperformed the S&P 500 benchmark in the U.S., most failed to generate positive annual returns in the Chinese market, underscoring the importance of evaluating models across multiple markets. Second, prediction accuracy was not strongly correlated with portfolio returns. The correlation between predicted and actual returns from the methods was too low to reliably indicate portfolio performance. Third, the advanced deep learning methods did not show better performance compared to traditional methods. More recently proposed methods based on graph neural network (GNN) and Variational autoencoder (VAE) did not demonstrate better performance. Finally, method performance was strongly influenced by the testing period. The top-performing methods varied depending on the timeframe used for evaluation, suggesting that previous studies, which typically test models over just 2-3 years (Deng et al., 2009; Du et al., 2021; Duan et al., 2022; Sawhney et al., 2021; Wang et al., 2022; Xia et al., 2024; Li et al., 2024), are too limited to definitively determine a model's superiority. Future research should aim for more comprehensive evaluations that assess both long-term and short-term performance.

The contributions of this work are summarized as follows:

- We created standard datasets across two distinct markets, reproduced stock prediction methods and integrated mainstream time-series forecasting methods into a benchmark for stock prediction.

- We comprehensively evaluated methods with long-term and short-term portfolio performance by simulating real-world trading scenarios, and revealed several key findings unreported from previous studies.

Overall, our research suggests that, unlike in other fields such as computer vision and natural language processing, machine learning has been slow to make progress in stock prediction. This presents a great opportunity for machine learning researchers to leverage our BenchStock for competition.

## 2 BACKGROUND

### 2.1 STOCK PREDICTION BACKGROUND

**Problem Formulation**  Stock prediction is commonly approached as a regression problem. Let us denote the set of stocks in the market as $S = \{s_1, s_2, \ldots, s_N\}$, where each stock is associated with observable features $X = \{x_1, x_2, \ldots, x_N\}$ and $x_i \in \mathbb{R}^d$. The objective is to learn a model $f$ that forecasts future returns $Y \in \mathbb{R}^N$, as formulated in the equation:

$$Y = f(X). \tag{1}$$

**Related Methods**  In asset pricing field, research mainly focuses on features $X$, commonly referred to as factors in finance, in the equation. Linear regression is prevalently used as the default model for testing the effectiveness of these factors. The representation of features $x_i$ has evolved across different models during the last few decades, which shifted from several factors in the early works (Sharpe, 1964; Fama & French, 1992) to hundreds of factors in more recent works (Cochrane, 2011; Mclean & Pontiff, 2016; Hou et al., 2018; Harvey et al., 2015). However, as more and more factors were being used, linear method started to struggle in learning patterns from high-dimensional data due to issues like collinearity. Luckily, with the emergence of artificial intelligence, researchers have begun to address these challenges using machine learning methods. A recent work constructed a dataset with 94 firm-level characteristics plus eight macroeconomic variables and applied multiple machine learning models to compare with linear method (Gu et al., 2020). The results showed that machine learning methods with ability to modeling non-linear relationship are generally more accurate.

Different from asset pricing studies (Sharpe, 1964; Fama & French, 1992; 2015), stock forecast research in machine learning domain emphasizes methodologies rather than the specific features used in the forecasting equations. Most of these works framed the task as a time-series forecasting problem and restricted their feature sets to trading price and volume. A comprehensive introduction to various machine learning methods for stock prediction is detailed in Sec. 3.3.

**Related Platforms**  For strict evaluation of stock prediction methods, a systematic pipeline of data processing, model training and backtesting is required. Despite the limited number of open-sourced platforms dedicated to stock prediction, there are a few noteworthy examples. FinRL-Meta (Liu et al., 2022) has compiled various datasets and established an environment for financial reinforcement learning tasks, setting a benchmark in RL-based methods. FinGPT (Wang et al., 2023) provides a platform for aggregating stock-related news and predicting stock movements using Large Language Models (LLMs).

Among various platforms, Qlib (Yang et al., 2020), developed by Microsoft, is an AI-oriented open-source quantitative investment tool that provides comprehensive functionalities for testing stock return prediction algorithms. It includes data processing, machine learning model prediction, and backtesting modules essential for real-world quantitative trading. However, there are several issues with Qlib that prevented us from using the platform, which we discussed in Appendix A.1. Consequently, we created our own datasets and framework for prediction methods, and only used Qlib's backtest testing module for evaluation.

## 3 *BenchStock*

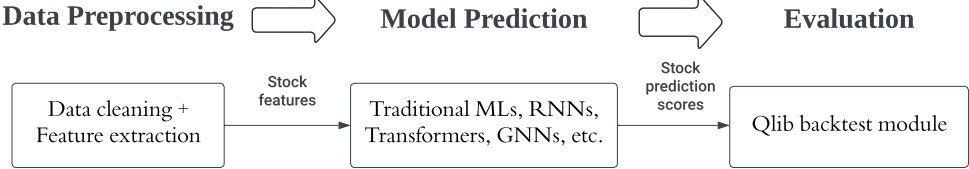

Figure 1: Overall framework of *BenchStock*

*BenchStock* is open source and free to use/modify under the MIT License. Since the stock datasets used in this benchmark are proprietary and require a subscription or institutional access, we provide a link to the data source along with a script for processing the data, rather than distributing the dataset directly. By running the provided script on the data downloaded from the link, a standardized dataset will be created. The overall framework of *Benchstock* is presented in Figure 1, and the implementation details will be introduced in the following sections.

### 3.1 DATASETS

Our benchmark features two price-volume-based datasets, each representing one of the world's largest stock markets, the United States and China, with daily frequency data. The data were fetched from authoritative data sources CRSP and CSMAR respectively. For each market, the daily open price, high price, low price, close price, trade volume and trade amount were used for prediction.

We focused on these two markets due to their significant differences in institutional backgrounds, which highlights the potential limitations of models when applied across different markets. The Chinese financial market is predominantly driven by retail investors, whereas the U.S. market is dominated by institutional investors. The Chinese market is also subject to more stringent trading regulations than the relatively liberal U.S. market. For example, the short trading has been strictly regulated in China for individual stocks, which does not exist in the U.S. market.

Unlike existing studies that typically focused on a single market (Gu et al., 2020; Deng et al., 2009; Du et al., 2021; Duan et al., 2022), our model comparison evaluated performance across different markets. Previous works typically divided individual markets into multiple datasets based on stock exchanges (Veličković et al., 2018; Sawhney et al., 2021; Wang et al., 2022). Our approach consolidates each market into a single dataset for a more comprehensive market overview. The U.S. market dataset includes securities from NYSE, NASDAQ, and AMEX, spanning from 1989 to 2023, while the China A-share market dataset encompasses stocks from both the Shanghai and Shenzhen Stock Exchanges, covering the period from 1990 to 2023. To make fair comparisons with methods based on graph neural network (GNN) in our benchmark (Sawhney et al., 2021; Wang et al., 2022; Xia et al., 2024; Li et al., 2024), which require consistent stock components throughout the dataset, we have also generated an auxiliary dataset containing only stocks that persisted throughout 2000-2023 and maintained a consistent number of trading days in each market. For clarity, we call the datasets covering all stocks "full datasets" (US-Full & CN-Full) and the auxiliary datasets containing consistent stocks "consistent datasets" (US-Con & CN-Con). The details of the datasets can be found in Appendix A.2.

### 3.2 PREPROCESSING

The data preprocessing of our benchmark include two main procedures: data cleaning and feature extraction. Specifically, we built a customized pipeline with strict procedures for cleaning data from each source, which includes removing noisy data, handling missing data and backward adjusting. The feature extraction procedure processed price and volume features into the same scale with differencing and normalization. The detail of preprocessing can be found in Appendix A.3.

### 3.3 STOCK PREDICTION METHODS

Our benchmark covers a variety of methods, spanning from traditional machine learning models to the latest deep-learning models in AI fields. For the convenience of discussion, we group these methods into five categories as follows:

- Traditional Machine Learning Methods: Linear Regression (LR), Gradient Boost (GBRT), Random Forest (RF), and Multilayer Perceptron (MLP) (Gu et al., 2020).
- RNN-Based Methods: LSTM (Hochreiter & Schmidhuber, 1997), DA-RNN (Deng et al., 2009), AdaRNN (Du et al., 2021).
- Transformer-Based Methods: Transformer (Vaswani et al., 2017), LogSparse (Li et al., 2019), Reformer (Kitaev et al., 2020), Informer (Zhou et al., 2021), Autoformer (Wu et al., 2021), Fedformer (Zhou et al., 2022), Crossformer (Zhang & Yan, 2023).
- GNN-based Methods: (GAT) (Veličković et al., 2018), STHAN-SR (Sawhney et al., 2021), ALSP-TF (Wang et al., 2022), CI-STHPAN (Xia et al., 2024), MASTER (Li et al., 2024).
- Other Methods: NLinear, DLinear (Zeng et al., 2023), FactorVAE (Duan et al., 2022), Mamba (Gu & Dao, 2024).

We treat the task as a time-series prediction problem, focusing exclusively on methods that forecast stock returns using price and volume factors. The information of methods in details are summarized in Appendix A.4.

Acknowledging the diversity of methods in this domain, we have reproduced models that do not have publicly released codes or are not designed for stock prediction. These reproductions, integrated into our PyTorch framework, were based on descriptions in the original papers. To support the research community, we commit to continuously updating our benchmark with novel, high-quality methods as they emerge. This ongoing effort aims to facilitate easy access to cutting-edge techniques in stock market analysis.

### 3.4 EVALUATION METHOD

We evaluated stock prediction results using a backtesting program from Microsoft's Qlib, which is designed to simulate real-world trading scenarios. Unlike traditional error-based metrics like MSE, Qlib assesses forecasts by forming portfolios and applying return-related financial metrics to gauge performance considering the purpose of model prediction is for portfolio management. For measuring the performance of the portfolio, we used financial metrics including Annual Return (AR), Sharpe Ratio (SR), Information Ratio (IR) and Maximum Drawdown (MDD). The explanation of metrics in detail is included in Appendix A.5. Higher values are preferable for all the metrics. For benchmark, we used S&P 500 and SSE Composite (SSEC) in the U.S. and the Chinese market respectively. The reason we used SSEC instead of normally selected CSI 300 is because CSI 300 started from 2005 and could not cover the whole sample period of our dataset.

To ensure a more comprehensive evaluation that closely reflects reality, we constructed portfolios based on daily forecasts and updated using a top-k strategy. We started with an initial capital of 100 million, maintaining 50 stocks and replacing $k = 10$ at the end of each trading day. Besides, we extended the evaluation period from the shorter spans common in prior research to three decades, providing insights into both long-term and short-term model performance. The implementation detail of our evaluation can be found in Appendix A.6.

## 4 EXPERIMENTS

### 4.1 TRAINING SETTING

In the stock dataset training process, a rolling approach was adopted where the training set spanned three years of data, and the validation set comprised two years of data. This approach involved iterative testing, where the model was trained on a subset of data for a specific time period, followed by validation on the subsequent time period, and finally tested on the next year's data. For example, starting with data from 1989 to 2023, the initial training set covered data from 1989 to 1991, while

| Methods | US-Full | | | | CN-Full | | | |
|---|---|---|---|---|---|---|---|---|
| | AR(%) | SR | IR | MDD(%) | AR(%) | SR | IR | MDD(%) |
| LR | 23.01 | 0.33 | 0.23 | -32.47 | -12.43 | -0.52 | -1.23 | -34.90 |
| GBRT | 23.13±4.78 | 0.39±0.07 | 0.28±0.05 | -30.73±1.20 | -14.99±2.42 | -0.62±0.08 | -1.51±0.16 | -35.56±1.06 |
| RF | 24.84±0.03 | 0.66±0.14 | 0.53±0.13 | -27.38±0.05 | -13.19±3.09 | -0.52±0.10 | -1.32±0.17 | -36.05±1.44 |
| MLP | 26.23±5.34 | 0.45±0.14 | 0.32±0.10 | -27.07±0.89 | -13.75±2.98 | -0.57±0.11 | -1.36±0.19 | -35.35±0.98 |
| LSTM | 40.65±4.01 | 0.63±0.25 | 0.53±0.21 | -26.60±0.91 | -12.06±2.87 | -0.51±0.10 | -1.22±0.17 | -34.44±1.16 |
| DA-RNN | 4.04±2.72 | 0.04±0.05 | -0.09±0.06 | -28.56±0.75 | -26.03±4.49 | -1.03±0.16 | -2.09±0.26 | -40.65±1.81 |
| AdaRNN | 18.78±4.23 | 0.39±0.12 | 0.26±0.08 | -18.56±0.27 | -4.32±0.92 | -0.24±0.03 | -0.88±0.07 | -29.28±0.48 |
| Transformer | 61.83±8.93 | 0.64±0.12 | 0.57±0.12 | -28.99±1.10 | -16.43±2.30 | -0.65±0.08 | -1.48±0.14 | -36.54±0.73 |
| Logsparse | 53.16±5.03 | 0.68±0.25 | 0.60±0.21 | -28.95±1.03 | -14.24±1.16 | -0.57±0.04 | -1.35±0.06 | -35.42±0.73 |
| Reformer | 48.77±7.42 | 0.71±0.26 | 0.62±0.24 | -28.28±1.07 | -13.04±1.57 | -0.53±0.06 | -1.34±0.09 | -35.34±0.86 |
| Informer | 53.90±6.97 | 0.80±0.00 | 0.71±0.01 | -27.76±0.43 | -10.36±1.54 | -0.44±0.05 | -1.16±0.10 | -33.99±0.90 |
| Autoformer | 36.73±4.33 | 0.36±0.20 | 0.29±0.16 | -26.48±1.52 | -9.64±0.96 | -0.42±0.04 | -1.17±0.07 | -33.92±0.23 |
| Fedformer | 27.14±4.49 | 0.56±0.08 | 0.42±0.05 | -25.12±1.91 | -13.07±0.93 | -0.53±0.03 | -1.30±0.04 | -35.50±0.90 |
| Crossformer | 44.61±4.07 | 0.82±0.29 | 0.69±0.23 | -27.23±1.24 | -10.39±2.53 | -0.45±0.09 | -1.12±0.16 | -34.02±1.05 |
| NLinear | 12.11±2.52 | 0.24±0.07 | 0.09±0.04 | -30.74±0.99 | -21.66±3.17 | -0.90±0.13 | -1.89±0.21 | -37.48±0.96 |
| DLinear | 5.82±5.29 | 0.04±0.06 | -0.09±0.09 | -32.35±0.78 | -15.24±0.32 | -0.63±0.01 | -1.42±0.03 | -36.09±0.15 |
| FactorVAE | 9.46±0.92 | 0.23±0.05 | 0.05±0.03 | -19.95±1.75 | 5.91±14.14 | 0.12±0.49 | -0.16±0.99 | -28.86±2.45 |
| Mamba | 15.59±3.21 | 0.21±0.09 | 0.12±0.07 | -29.83±0.63 | -15.54±0.78 | -0.65±0.03 | -1.50±0.05 | -35.53±0.19 |
| Benchmark | 8.06 | 0.32 | - | -14.97 | 6.91 | 0.20 | - | -22.57 |

Table 1: Evaluation results from 1994 to 2023 on the US-Full dataset and from 1996 to 2023 on the CN-Full dataset. The results are in format of (average ± standard deviation) for 5 trials. annual return and maximum drawdown are shown in percentage. The benchmark for US-Full and CN-Full datasets in this table denotes S&P 500 and SSEC index respectively. The names of the methods can be found in Sec 3.3.

the validation set included data from 1992 to 1993. The model parameters with the best accuracy on the validation set were selected to predict the test set, which included data from 1994. In the next iteration, the training set slid forward to cover data from 1990 to 1992, while the validation set shifted to data from 1993 to 1994, and testing was conducted on data from 1995. This sliding process ensured that the model adapted to changing market dynamics year by year. As a result, the U.S. market data was tested from 1994 to 2023, and the Chinese market was evaluated from 1996 to 2023, as most of its components started in 1991.

We used the past 64 trading days' feature sequence as input and the next trading day's return as label. For GNN-based methods that require prior knowledge of stock relationships (Veličković et al., 2018; Sawhney et al., 2021), we used industry information from CRSP and CSMAR databases to assume stocks within the same industry as neighbors. This approach is adopted due to the absence of text data like news. All methods were run 5 times from different random initial points [1]. The details for the hyperparameters used for training each method are summarized in Appendix A.7.

## 4.2 Results and Analysis

In this section, we present the evaluation results for the U.S. and the Chinese markets and analyze the outcomes. The results in tables are average value and standard deviation of 5 trials. To demonstrate more details, we also draw average log(cumulative value) of portfolios from 5 trials for all methods. From the comparison of methods across four datasets in two different markets, we have some interesting observations.

**Most methods demonstrated strong performances in the U.S. market.** All methods demonstrated positive annualized returns (ARs) and Sharpe Ratios (SRs) over the U.S. market, which are shown in Table 1. All but two methods managed to achieve positive Information Ratios (IRs), meaning they outperformed the benchmark S&P 500 index. The consistent growth from market itself apparently contributed to the strong performances of machine learning methods. Except for the two major market crises, the bursting of the dot-com bubble in 2000-2002 and the financial crisis in 2008, S&P 500 rarely experienced years with negative returns. From Appendix A.8, we can observe such stability from cumulative value curves of most methods. Despite the overall impressive

---

[1] For methods (Transformer, Informer) that took too long to run for a single trial, we used the forecast results from the last five epochs of each year's model as the outcomes for five trials. For Linear Regression, we only ran it for once since the optimal parameter is fixed.

| Methods | US-Full | | | | US-Con | | | |
|---|---|---|---|---|---|---|---|---|
| | AR(%) | SR | IR | MDD(%) | AR(%) | SR | IR | MDD(%) |
| LR | 3.59 | 0.03 | -0.07 | -33.84 | 18.26 | 0.56 | 0.62 | -23.55 |
| GBRT | 6.29±4.24 | 0.08±0.09 | -0.05±0.12 | -30.56±1.04 | 20.52±0.99 | 0.62±0.04 | 0.71±0.06 | -23.82±0.40 |
| RF | 6.30±0.57 | 0.15±0.02 | -0.05±0.02 | -26.60±0.01 | 23.06±0.92 | 0.68±0.03 | 0.82±0.05 | -24.07±0.41 |
| MLP | 4.17±4.64 | 0.01±0.09 | -0.13±0.13 | -27.21±0.54 | 15.50±1.48 | 0.48±0.05 | 0.48±0.08 | -23.03±1.28 |
| LSTM | 13.55±1.82 | 0.37±0.05 | 0.20±0.06 | -27.19±0.69 | 20.80±0.51 | 0.64±0.02 | 0.75±0.03 | -21.85±0.76 |
| DA-RNN | -12.19±4.33 | -0.53±0.20 | -0.78±0.23 | -30.98±1.21 | 12.07±1.17 | 0.40±0.03 | 0.33±0.07 | -20.96±0.66 |
| AdaRNN | 5.20±1.13 | 0.16±0.05 | -0.22±0.11 | -18.14±0.14 | 14.79±1.71 | 0.50±0.07 | 0.57±0.13 | -19.88±0.33 |
| Transformer | 61.83±8.93 | 0.64±0.12 | 0.57±0.12 | -28.99±1.10 | 15.85±1.20 | 0.45±0.04 | 0.43±0.07 | -24.94±0.21 |
| LogSparse | 17.60±4.79 | 0.39±0.17 | 0.25±0.14 | -30.36±0.92 | 21.14±2.69 | 0.62±0.08 | 0.71±0.12 | -24.36±0.41 |
| Reformer | 15.97±7.45 | 0.43±0.22 | 0.27±0.24 | -29.57±1.68 | 21.11±1.74 | 0.63±0.06 | 0.74±0.11 | -24.33±0.77 |
| Informer | 53.90±6.97 | 0.80±0.00 | 0.71±0.01 | -27.76±0.43 | 19.80±1.16 | 0.57±0.04 | 0.62±0.06 | -24.93±0.23 |
| Autoformer | 10.27±6.21 | 0.15±0.11 | 0.01±0.15 | -26.39±3.05 | 13.31±2.48 | 0.43±0.07 | 0.39±0.16 | -22.02±2.09 |
| Fedformer | 4.51±2.54 | 0.11±0.11 | -0.11±0.10 | -25.17±2.69 | 13.09±1.78 | 0.36±0.05 | 0.30±0.09 | -24.59±0.88 |
| Crossformer | 23.33±5.20 | 0.65±0.23 | 0.49±0.20 | -27.17±2.35 | 20.43±1.00 | 0.59±0.03 | 0.66±0.05 | -25.01±0.86 |
| GAT | - | - | - | - | 13.56±3.76 | 0.44±0.12 | 0.39±0.26 | -20.99±1.82 |
| STHAN-SR | - | - | - | - | 11.58±3.34 | 0.41±0.14 | 0.38±0.31 | -18.05±0.95 |
| ALSP-TF | - | - | - | - | 10.57±1.58 | 0.38±0.05 | 0.28±0.14 | -18.68±1.86 |
| CI-STHPAN | - | - | - | - | 10.64±3.13 | 0.33±0.11 | 0.19±0.21 | -21.02±1.57 |
| MASTER | - | - | - | - | 6.04±0.09 | 0.26±0.01 | -0.12±0.01 | -14.26±0.06 |
| NLinear | 0.61±3.86 | -0.04±0.12 | -0.21±0.12 | -32.09±0.95 | 14.09±0.61 | 0.40±0.02 | 0.37±0.03 | -22.94±0.30 |
| DLinear | -7.22±2.55 | -0.26±0.12 | -0.44±0.16 | -34.44±1.35 | 16.24±0.82 | 0.55±0.02 | 0.57±0.03 | -22.06±0.39 |
| FactorVAE | 5.16±2.20 | 0.13±0.09 | -0.11±0.07 | -20.90±1.10 | -1.72±1.83 | -0.16±0.08 | -0.91±0.22 | -20.34±0.83 |
| Mamba | -0.81±2.41 | -0.10±0.09 | -0.27±0.13 | -32.35±0.68 | 20.61±0.50 | 0.65±0.02 | 0.76±0.03 | -22.40±0.18 |
| S&P 500 | 7.48 | 0.28 | - | -15.00 | 7.48 | 0.28 | - | -15.00 |

Table 2: Evaluation results from 2005 to 2023 on the US-Full and the US-Con dataset. The results are in format of (average ± standard deviation) for 5 trials. annual return (AR) and maximum drawdown (MDD) are shown in percentage. The names of the methods can be found in Sec 3.3.

| Methods | CN-Full | | | | CN-Con | | | |
|---|---|---|---|---|---|---|---|---|
| | AR(%) | SR | IR | MDD(%) | AR(%) | SR | IR | MDD(%) |
| LR | -14.46 | -0.61 | -1.16 | -35.57 | -10.59 | -0.46 | -0.98 | -34.05 |
| GBRT | -17.66±3.39 | -0.72±0.12 | -1.43±0.20 | -36.96±1.37 | -10.45±0.85 | -0.45±0.03 | -1.03±0.06 | -33.55±0.23 |
| RF | -15.78±4.13 | -0.61±0.13 | -1.28±0.22 | -36.77±1.94 | -13.53±1.27 | -0.53±0.04 | -1.15±0.08 | -35.74±0.63 |
| MLP | -16.05±3.78 | -0.66±0.14 | -1.25±0.22 | -36.61±1.23 | -5.88±1.04 | -0.31±0.04 | -0.71±0.07 | -31.48±0.62 |
| LSTM | -15.26±3.17 | -0.62±0.11 | -1.16±0.17 | -36.14±1.53 | -8.73±0.55 | -0.43±0.02 | -0.91±0.04 | -30.11±0.24 |
| DA-RNN | -31.42±5.54 | -1.22±0.20 | -2.10±0.29 | -43.82±2.60 | -24.17±0.47 | -0.94±0.02 | -1.77±0.03 | -39.05±0.20 |
| AdaRNN | 1.49±1.11 | -0.02±0.04 | -0.25±0.09 | -26.40±0.34 | -13.63±1.21 | -0.56±0.05 | -1.17±0.07 | -33.40±0.49 |
| Transformer | -18.59±2.38 | -0.72±0.09 | -1.35±0.13 | -38.36±0.97 | -6.54±0.27 | -0.31±0.01 | -0.72±0.02 | -33.35±0.16 |
| LogSparse | -15.87±1.16 | -0.63±0.04 | -1.21±0.06 | -36.42±0.63 | -9.63±0.45 | -0.41±0.01 | -0.91±0.03 | -34.60±0.18 |
| Reformer | -14.24±1.74 | -0.58±0.06 | -1.16±0.09 | -36.11±0.94 | -7.65±0.30 | -0.34±0.01 | -0.83±0.02 | -33.07±0.12 |
| Informer | -11.45±1.92 | -0.48±0.07 | -1.01±0.12 | -34.81±1.08 | -4.09±0.48 | -0.23±0.02 | -0.61±0.03 | -29.80±0.27 |
| Autoformer | -10.87±1.86 | -0.47±0.07 | -0.99±0.12 | -34.40±0.16 | -7.34±0.25 | -0.36±0.01 | -0.80±0.04 | -32.67±0.15 |
| Fedformer | -14.30±1.00 | -0.58±0.04 | -1.14±0.05 | -36.15±0.90 | -10.52±0.64 | -0.45±0.02 | -0.94±0.04 | -34.28±0.20 |
| Crossformer | -13.60±2.33 | -0.58±0.08 | -1.11±0.13 | -35.93±1.38 | -0.21±0.77 | -0.08±0.03 | -0.33±0.06 | -29.69±0.31 |
| GAT | - | - | - | - | -17.22±15.66 | -0.68±0.57 | -1.34±1.00 | -36.01±5.97 |
| STHAN-SR | - | - | - | - | -4.84±9.42 | -0.23±0.34 | -0.61±0.63 | -31.32±3.63 |
| ALSP-TF | - | - | - | - | 1.24±5.59 | -0.01±0.23 | -0.23±0.34 | -27.42±6.84 |
| CI-STHPAN | - | - | - | - | -10.52±0.64 | -0.45±0.02 | -0.94±0.04 | -34.28±0.20 |
| MASTER | - | - | - | - | -1.67±0.28 | -0.13±0.01 | -0.43±0.02 | -31.08±0.12 |
| NLinear | -24.81±3.18 | -1.03±0.14 | -1.78±0.18 | -39.32±1.17 | -11.01±0.24 | -0.46±0.01 | -0.95±0.02 | -33.24±0.16 |
| DLinear | -17.67±0.38 | -0.71±0.01 | -1.30±0.03 | -37.50±0.28 | -14.05±0.61 | -0.59±0.02 | -1.18±0.05 | -34.14±0.45 |
| FactorVAE | 8.22±17.65 | 0.19±0.61 | 0.09±1.15 | -28.22±2.55 | 10.23±10.01 | 0.31±0.37 | 0.35±0.66 | -26.99±2.39 |
| Mamba | -24.88±0.47 | -0.99±0.02 | -1.71±0.03 | -39.51±0.33 | -13.68±0.44 | -0.57±0.01 | -1.18±0.03 | -35.06±0.28 |
| SSEC | 4.59 | 0.11 | - | -22.54 | 4.59 | 0.11 | - | -22.54 |

Table 3: Evaluation results from 2005 to 2023 on CN-Full and CN-Con dataset. The results are in format of (average ± standard deviation) for 5 trials. annual return (AR) and maximum drawdown (MDD) are shown in percentage. The names of the methods can be found in Sec 3.3.

results, the US-Full dataset revealed significant variations in performance among different methods. While traditional machine learning methods and Transformer-based methods clearly outperformed the benchmark index, RNN-based methods (except LSTM) and other methods demonstrated inferior performances compared to traditional machine learning approaches. DA-RNN and DLinear even underperformed the benchmark S&P 500. The more advanced Transformer-based methods appeared to offer superior performance. Three Transformer-based methods achieved Sharpe ratios and information ratios around 0.7, suggesting their efficacy in capturing the complexities of the US-Full market dynamics. The dominance of Transformer-based methods aligns with their proven superiority in general time-series forecasting tasks from their original works (Vaswani et al., 2017; Li et al., 2019; Kitaev et al., 2020; Zhou et al., 2021; Wu et al., 2021; Zhang & Yan, 2023). It is also against the claim that Transformer-based methods are less effective than the two linear methods (NLinear & DLinear) in time-series forecasting (Zeng et al., 2023). These findings highlight the advantage of model sophistication and adaptability in achieving competitive results over the U.S. market.

**No methods performed well in the Chinese Market.** Unlike their performance in the U.S. market, almost all methods failed to generate positive returns in the Chinese market. As shown in Table 1, only FactorVAE (Duan et al., 2022) achieved positive AR and SR on the CN-Full dataset. Apart from FactorVAE, portfolios from other methods lost at least 75% of their initial capital over the 28-year testing period. These poor performances in the Chinese market can be attributed to several factors, with market volatility being a significant one. While the U.S. market consistently reached new highs, the Chinese market fluctuated severely and remained stagnant for over a decade. Consequently, most methods struggled with this volatility, and their portfolio values continued to decline under the long-only top-k strategy. Additionally, difference in trading cost contributed to the bad performance of machine learning methods in the Chinese market. Commission fees in the Chinese market are based on transaction value, making daily trading strategies significantly more expensive in China compared to the U.S. As observed in Appendix A.9 and Figure 2, although the mean returns from most methods' portfolios in the Chinese market were positive, they were insufficient to cover the trading costs (0.25% for sell orders and 0.15% for buy orders). These findings underscore the necessity of testing models in various markets, as success in one does not guarantee success in another. It also highlights the necessity of testing stock prediction methods in real-world scenarios, as a method that does not account for trading costs may report positive returns while actually resulting in a loss. As a result, future research in the Chinese market should focus on strategies to reduce trading costs in addition to improving prediction accuracy.

**The prediction accuracy was not correlated with portfolio returns.** One might expect that better predictions of stock prices would lead to higher returns. To evaluate the accuracy of predictions for stocks selected into portfolios, we measured it using the information coefficient (IC), which ranges from -1 to 1 and represents the correlation between predicted and actual returns. The results, shown in Figure 2 and Appendix A.9, indicate that prediction accuracy did not strongly correlate with the mean return of selected stocks, particularly in the Chinese market. Although methods with higher prediction accuracy generally yielded higher mean returns, some methods with ICs close to 0 also achieved similar mean returns. Specifically, while methods like GBRT, RF, LSTM, and most Transformer-based models demonstrated better predictive ability, this did not translate to superior returns compared to methods with lower ICs. Surprisingly, the IC in the Chinese market was actually better than in the U.S. market, especially on consistent datasets.

The predictability of stock data is quite low due to its inherently low signal-to-noise ratio. Most ICs are typically below 0.1, which is insufficient to guarantee higher returns when a method's IC is only slightly better than others. With such low predictability, portfolio performance largely depended on market conditions. In a stable market like the U.S., methods benefited from continuous market growth. However, in the more volatile Chinese market—where 12 of the past 23 years experienced annual losses—methods suffered from market fluctuations. This explains why methods performed worse in the Chinese market despite demonstrating better prediction accuracy. Consequently, improvements in current model prediction accuracy were not significant enough to ensure better portfolio returns.

**Advanced deep learning methods did not show obvious advantage to traditional methods.** Surprisingly, our results showed that advanced deep learning methods for stock prediction did not outperform traditional machine learning methods. First, we examined recently proposed advanced

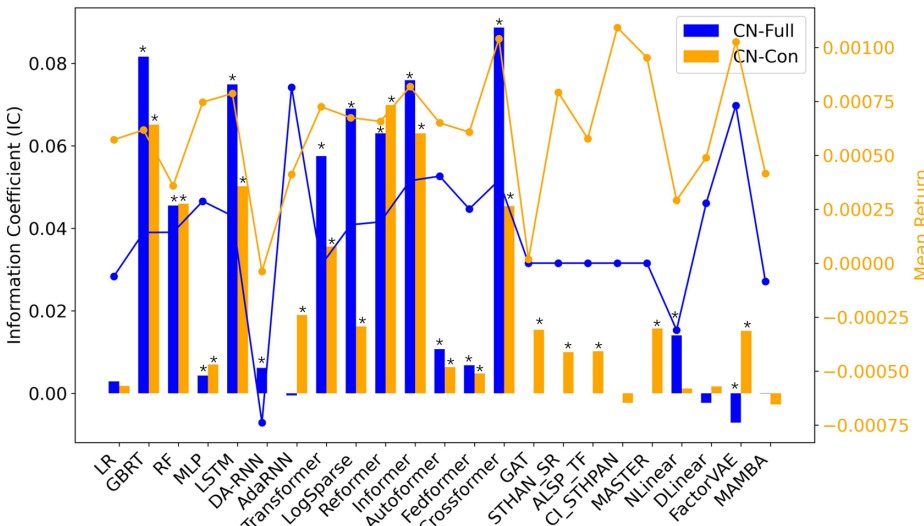

Figure 2: Comparison of IC between full dataset and consistent dataset in the Chinese market. The bars in the figure represent ICs and the dots represent mean returns of stocks selected in the portfolios. The "*" sign in Figures 2 represents the IC of the method is statistically significant different from 0 at 0.05 level on the corresponding dataset. GNN-based methods have zero values for IC and return on full datasets because they were only tested on consistent datasets.

machine learning methods for stock prediction including GNN-based methods and FactorVAE. Contrary to our expectation, GNN-based approaches, as illustrated in Tables 2 and 3, did not outperform other types of methods on consistent datasets across both markets. In the U.S. market, they were among the least effective, with their only advantage being reduced portfolio volatility through maximum drawdowns (MDDs). However, their return-related metrics—AR, SR, and IR—were all lower than those of other methods. While ALSP-TF and MASTER performed relatively better than other methods in the Chinese market, their negative SR and AR indicate that they not only underperformed the market but also failed to surpass the risk-free rate. Furthermore, as shown in Appendix A.9 and Figure 2, GNN prediction accuracy was noticeably lower than that of other approaches. The differences between our findings and the literature may stem from varying evaluation methodologies, as our long-term top-k strategy contrasts with the single-year buy/sell strategies in prior studies. Our implementation of ALSP-TF (Wang et al., 2022) relied solely on our understanding due to a lack of provided code, and we also adjusted the STHAN-SR implementation, which originally selected optimal parameters based on test rather than validation metrics. Among the advanced stock prediction methods, only FactorVAE (Duan et al., 2022) achieved positive AR and SR in both markets, yet it ranked poorly in the U.S. and showed considerable instability between trials in the Chinese market, suggesting its results fluctuated between randomly good and poor, as detailed in Figure 2. As noted in Sec. 4.2, the prediction accuracy is insufficient to directly determine returns, and the higher returns from FactorVAE likely result from randomness rather than genuine predictive capability.

Second, we examined advanced machine learning methods not originally designed for stock prediction, including Transformer-based approaches that excel in time-series forecasting. However, these methods did not achieve similar success in stock prediction. Although they demonstrated better prediction accuracy, the improvements were too minimal to enhance portfolio performance, leading to strong results in the stable U.S. market but struggles in the more volatile Chinese market. Similarly, Mamba, a recent method proposed to improve on RNN and Transformer approaches, failed to deliver the expected advantages.

One possible reason for the underperformance of these advanced methods is that stock data significantly differs from other data types with stronger signals, and varying market regulations further complicate the situation. It appears that effective stock market prediction methods should prioritize improved data preprocessing and trading strategies rather than merely increasing model complexity.

| Methods | US-Full | | | | CN-Full | | | |
|---|---|---|---|---|---|---|---|---|
| | AR(%) | SR | IR | MDD(%) | AR(%) | SR | IR | MDD(%) |
| LR | -4.09 | -0.15 | -0.37 | -43.06 | -28.90 | -1.55 | -1.79 | -32.64 |
| GBRT | 0.39±8.48 | -0.09±0.21 | -0.32±0.27 | -40.98±3.37 | -31.68±3.58 | -1.83±0.15 | -2.33±0.24 | -34.52±3.14 |
| RF | 4.85±0.33 | 0.09±0.01 | -0.18±0.01 | -32.15±0.36 | -27.04±2.52 | -1.57±0.07 | -2.13±0.15 | -30.93±1.63 |
| MLP | -5.83±6.51 | -0.33±0.28 | -0.63±0.30 | -32.76±2.85 | -24.24±4.47 | -1.34±0.22 | -1.63±0.30 | -30.37±2.42 |
| LSTM | -4.67±2.47 | -0.20±0.07 | -0.51±0.09 | -38.41±1.72 | -25.34±2.96 | -1.31±0.09 | -1.61±0.17 | -32.65±2.11 |
| DA-RNN | -28.63±7.77 | -0.94±0.30 | -1.31±0.37 | -46.55±6.91 | -43.76±3.56 | -2.07±0.06 | -2.58±0.09 | -45.17±3.52 |
| AdaRNN | 3.06±1.82 | 0.04±0.07 | -0.57±0.17 | -22.06±0.90 | -1.25±0.51 | -0.18±0.03 | -0.06±0.05 | -16.12±0.09 |
| Transformer | 7.60±6.83 | 0.13±0.17 | -0.09±0.21 | -43.37±1.85 | -32.20±1.17 | -1.59±0.07 | -1.98±0.09 | -37.19±0.83 |
| Logsparse | 5.82±3.49 | 0.10±0.09 | -0.12±0.09 | -41.28±1.43 | -24.89±0.83 | -1.25±0.05 | -1.54±0.05 | -32.13±0.57 |
| Reformer | 4.03±6.21 | 0.05±0.16 | -0.18±0.18 | -39.80±2.46 | -26.51±2.68 | -1.40±0.14 | -1.79±0.19 | -31.60±1.56 |
| Informer | 8.06±3.04 | 0.13±0.07 | -0.05±0.09 | -39.42±1.91 | -24.56±2.21 | -1.27±0.11 | -1.54±0.13 | -31.21±1.68 |
| Autoformer | 3.40±9.75 | 0.00±0.34 | -0.32±0.38 | -28.40±4.11 | -18.08±1.05 | -0.99±0.05 | -1.22±0.09 | -25.71±1.27 |
| Fedformer | -5.48±8.80 | -0.34±0.43 | -0.77±0.48 | -26.32±6.40 | -27.74±1.98 | -1.41±0.11 | -1.81±0.12 | -33.39±0.89 |
| Crossformer | 32.85±7.01 | 0.74±0.30 | 0.58±0.26 | -30.00±6.42 | -27.48±4.12 | -1.48±0.16 | -1.92±0.25 | -33.93±3.27 |
| NLinear | -1.66±15.72 | -0.18±0.55 | -0.43±0.56 | -29.47±2.63 | -31.49±1.64 | -1.66±0.06 | -1.91±0.06 | -36.81±1.19 |
| DLinear | -20.07±11.62 | -0.61±0.32 | -0.93±0.35 | -47.33±7.08 | -31.27±1.05 | -1.57±0.04 | -1.76±0.07 | -36.19±0.80 |
| FactorVAE | 0.72±5.01 | -0.02±0.11 | -0.28±0.07 | -39.08±3.87 | 3.12±18.47 | -0.12±0.95 | 0.01±1.38 | -18.27±1.16 |
| Mamba | -10.14±3.62 | -0.30±0.15 | -0.51±0.22 | -44.58±2.06 | -26.52±1.11 | -1.33±0.06 | -1.51±0.07 | -32.58±0.87 |
| Benchmark | 10.23 | 0.36 | - | -18.70 | -0.62 | -0.16 | - | -14.72 |

Table 4: Evaluation results since COVID-19 over two markets. Annual return and maximum drawdown were shown in percentage. The names of the methods can be found in Sec 3.3.

**The performances of methods strongly depended on the testing period.** The performances of the methods were significantly influenced by the testing period, with the best method varying depending on the specific sample period used. Most methods achieved high annualized return due to great performances before 2000 in the U.S. market. To analyze their performances afterwards, we summarized the performances over two other periods, one since 2008 financial crisis and another since COVID-19 in Appendix A.10 and Table 4 respectively. The metrics of methods varied a lot, especially in the U.S. market. Despite being dominant during the 30-year evaluation period, most Transformer-based methods had less advantages in more recent years. In fact, when testing on the most recent 4 years since COVID-19, Fedformer even generated negative return and only Crossformer kept a relatively high annual return around 30% in the U.S. market. Similarly, in the Chinese market, most methods' performances deteriorated when being tested under more recent periods. Such differences are reflection of the market performance and macro economic condition, as most economies struggled during the COVID-19 era. After all, evaluation in both long-term and short-term can better examine a model's stability under different market conditions. Previously methods (Deng et al., 2009; Du et al., 2021; Duan et al., 2022; Veličković et al., 2018; Sawhney et al., 2021; Wang et al., 2022; Li et al., 2024) tested only for 2-3 years could lead to biased conclusions. As a result, we believe the future research should all test both long-term and short-term performances of methods for a more comprehensive evaluation.

## 5 DISCUSSION

In this paper, we introduce a benchmark for evaluating machine learning methods in forecasting stock movements. We collected and strictly preprocessed two datasets from the U.S. and Chinese stock markets, implemented 23 methods, and evaluated them through a unified backtesting program. The evaluation spanned three decades, offering insights into both long-term and short-term performance for comprehensive assessments. We analyzed the results and revealed findings that could hopefully provide insights for future stock prediction research.

Despite evaluating stock prediction methods in a comprehensive way, our study has several limitations. First, the reproduction of the benchmark methods using PyTorch was based on our understanding of the original works. There may be differences in implementation details that led to varying results. Second, this work purely focused on prediction methods using basic price and volume data. In the future, we plan to address these issues by improving datasets with additional features, and potentially incorporating Large Language Models (LLMs) for more accurate and reliable stock market forecasting.

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

# A APPENDIX

## A.1 PROBLEM WITH QLIB

Despite perfectly fitting the functional requirements for benchmarking stock return prediction methods, there are several issues within the platform. First, Qlib's built-in data collection script downloads data from Yahoo Finance, whose records exclude delisted stocks' information. This leads to a problem known as survivorship bias (Brown et al., 1992) in finance, which means that performance would become better if an truncated stock pool is used. Therefore, we replaced Qlib's data source to better align with actual market scenarios. Additionally, Qlib's normal training methods are limited to the traditional machine learning approach, where the training, validation, and test sets are fixed. This approach is not ideal for financial data due to the frequent changes in stock market's statistical distribution. Using a model trained on stock data from 2000 to 2008 to predict stock returns from 2010 to 2020 is evidently not an reasonable strategy. Although Qlib has added a rolling training mode that allows users to train with changing datasets, this function is still relatively restricted. The initial training date in rolling mode is fixed, which means the training set grows larger as it progresses to test more recent data. This becomes an issue when testing over long-term data, as a test set with 30 years of data means the training set would eventually include all 30 years of data. Thirty years of training data in DataFrame format would take over 300 gigabytes of memory and include many stocks that no longer exist in the market. The model would then likely fit delisted stocks and overfit historical data. In this work, we chose to use a rolling training mode that only included the most recent five years of data in our training and validation sets and discarded outdated data. With this consideration, we decided to create our own datasets and prediction methods under this training framework and incorporate only Qlib's backtesting module for evaluation.

## A.2 DATASET

| Market | #Stocks | Period |
|--------|---------|-----------|
| US-Con | 1057 | 2000-2023 |
| US-Full | 29019 | 1989-2023 |
| CN-Con | 748 | 2000-2023 |
| CN-Full | 3405 | 1990-2023 |

Table 5: Details of the U.S. and the Chinese market datasets.

The statistics of the four datasets are summarized in Table 5. The full U.S. dataset comprises 29,019 stocks, with 1,057 of them existing throughout the entire duration from 2000 to 2023 in consistent dataset. The full China A-share dataset includes 3,405 stocks, with 748 persisting from 2000 to 2023. Notably, the China A-share dataset is confined to stocks from the mainboard and SME (small and medium-sized enterprises) board[3], excluding other boards with investor experience requirements and higher daily price limits due to trade regulation differences.

A year-by-year analysis of each market's composition, detailed in Table 6, reveals that the U.S. market experienced more frequent changes in its components. Hundreds of stocks entered and exited the market each year from 2000 to 2023, in contrast to the Chinese market, where only a few companies left the market over the past decade. The reason for including only stocks after 2000 in the auxiliary dataset is to ensure there are enough stocks in the pool. If we selected stocks continually traded from 1990 to 2023, there would be too few stocks left. For clarity, we call the datasets covering all stocks "full datasets" (US-Full & CN-Full) and the auxiliary datasets containing consistent stocks "consistent datasets" (US-Con & CN-Con).

---

[3]The mainboard and SME board had same regulations for trading and requirements for investors, and the only difference comes from size of companies in the boards. By 2021, the SME board was incorporated into mainboard.

| Year | US | | | China | | |
|------|------|------|-------|-------|------|-------|
|      | Enter | Quit | Total | Enter | Quit | Total |
| 1989 | 460 | 621 | 5608 | 0 | 0 | 0 |
| 1990 | 452 | 562 | 5443 | 6 | 0 | 6 |
| 1991 | 600 | 508 | 5581 | 7 | 0 | 13 |
| 1992 | 812 | 619 | 7388 | 40 | 0 | 53 |
| 1993 | 1149 | 367 | 8111 | 124 | 1 | 176 |
| 1994 | 937 | 496 | 8700 | 111 | 0 | 287 |
| 1995 | 883 | 616 | 9076 | 24 | 0 | 311 |
| 1996 | 1202 | 644 | 9678 | 201 | 0 | 512 |
| 1997 | 883 | 815 | 9922 | 208 | 0 | 720 |
| 1998 | 685 | 1073 | 9789 | 106 | 1 | 825 |
| 1999 | 763 | 1047 | 9453 | 98 | 0 | 923 |
| 2000 | 777 | 1027 | 9163 | 132 | 0 | 1055 |
| 2001 | 315 | 978 | 8442 | 84 | 0 | 1139 |
| 2002 | 303 | 701 | 7744 | 71 | 4 | 1206 |
| 2003 | 270 | 599 | 7312 | 67 | 8 | 1265 |
| 2004 | 462 | 450 | 7181 | 101 | 5 | 1361 |
| 2005 | 481 | 486 | 7243 | 15 | 11 | 1365 |
| 2006 | 549 | 464 | 7320 | 63 | 13 | 1415 |
| 2007 | 730 | 602 | 7600 | 127 | 27 | 1515 |
| 2008 | 321 | 544 | 7342 | 80 | 19 | 1576 |
| 2009 | 294 | 507 | 7082 | 72 | 7 | 1641 |
| 2010 | 479 | 459 | 7055 | 234 | 13 | 1862 |
| 2011 | 471 | 392 | 7083 | 167 | 9 | 2020 |
| 2012 | 415 | 451 | 7095 | 87 | 6 | 2101 |
| 2013 | 504 | 373 | 7159 | 17 | 3 | 2115 |
| 2014 | 626 | 361 | 7435 | 71 | 5 | 2181 |
| 2015 | 585 | 410 | 7664 | 137 | 5 | 2313 |
| 2016 | 455 | 524 | 7699 | 152 | 6 | 2459 |
| 2017 | 571 | 476 | 7764 | 299 | 7 | 2751 |
| 2018 | 667 | 471 | 7957 | 83 | 4 | 2830 |
| 2019 | 576 | 480 | 8065 | 84 | 8 | 2906 |
| 2020 | 829 | 464 | 8413 | 146 | 12 | 3040 |
| 2021 | 1718 | 409 | 9672 | 123 | 13 | 3150 |
| 2022 | 828 | 637 | 10121 | 74 | 18 | 3206 |
| 2023 | 802 | 639 | 10284 | 58 | 34 | 3230 |

Table 6: The number of stocks entered and quit the two markets in each year.

## A.3  DATA PREPROCESSING

### A.3.1  DATA CLEANING

Data cleaning is especially important for stock data since different data sources have various formats and include lots of missing data. Even the most authoritative data sources have errors, so we built a customized pipeline with strict procedures for handling data from each source. First, we removed data that could bring extra noise in training, including duplicated data, data on IPO date of each stock and data of stocks being suspended. This assures all samples in dataset are valid trading records. Second, missing data were handled differently in various cases. Data with too little information were removed. For instance, if one sample had no volume-related data or missed all price-related data, we filtered them to avoid creating extremely inaccurate information. For samples with few price features missed, we filled them with other price features from the same sample. Third, the whole dataset was backward adjusted to eliminate effects from events like stock split and dividends. This procedure is highly important as any unprocessed stock split would fully change the returns from the stock. In the end, features from different data sources are converted into same format for training under the unified framework.

### A.3.2 FEATURE EXTRACTION

For features including open, high, low, and close prices, trade volume, and trade amount, two steps were implemented to process them into same scale. First, raw price and volume data were transformed into percentage change. Specifically, high, low, close prices' were processed into percentage change from open price, and open price was processed into difference from previous day's close price. For trade volume and amount, the first difference of the time-series were calculated. Close price's first difference was also included in addition to difference from open price. After the first step, all features were normalized with two steps: centering and scaling. During centering step, the median was subtracted from each feature, shifting the median of the feature to zero. This step ensures that the central tendency of the feature is not influenced by outliers. Following centering, the feature values were scaled by dividing the interquartile range (IQR), which is the difference between the 75th percentile and the 25th percentile of the feature. This scaling step normalizes the spread of the feature values while being robust to outliers, as the IQR is a measure of statistical dispersion that is not affected by extreme values. After all the procedures, the features were processed into a uniform scale.

### A.4 METHODS

The methods included in *BenchStock* can be summarized as the following types:

**I. Traditional Machine Learning Methods**  Traditional machine learning methods are those previously used in empirical asset pricing fields, including Linear Regression (LR), Gradient Boost (GBRT), Random Forest (RF), and Multilayer Perceptron (MLP) (Gu et al., 2020). For decades, linear regression has been the default method in traditional asset pricing field. Researchers used linear regression to explore the effectiveness of predictors in explaining stock returns. However, as more and more predictors being tested, it became too demanding for linear regression to handle such high-dimensional data. The inability to simulate non-linear relationships and address collinearity prevents it from being the ideal method for asset pricing in the big data era. As a result, machine learning methods are applied to avoid the restrictions of linear models. Experiments showed that GBRT, RF and MLP demonstrated better performances compared to linear regression when using 94 firm-level predictors (Gu et al., 2020). We used these traditional methods as the baselines and examined whether the recent advanced AI models would outperform them.

**II. RNN-Based Models**  Recurrent Neural Network (RNN) is renowned for its ability to handle sequential data, which makes it an ideal candidate for predicting stock returns. We included LSTM (Hochreiter & Schmidhuber, 1997), the most widely applied variation of the vanilla RNN method, as our baseline. In addition, we adopted several variation of RNN methods designed for stock return prediction. A typical method is DA-RNN (Deng et al., 2009), which adds attention mechanism to RNN network. Another RNN-based method is AdaRNN (Du et al., 2021), which incorporates adaptive learning into stock prediction. Due to the time-varying conditional distribution of stock data, the distribution of test data could be significantly different from the training dataset. To mitigate the overfitting issue caused by a significant distribution shift between training and testing data, AdaRNN applies techniques that divide training sets into different groups, enabling better generalization on unseen data.

**III. Transformer-Based Methods**  Transformer (Vaswani et al., 2017) has revolutionized machine learning, significantly impacting both natural language processing and computer vision. This prominence extends to long-term time-series forecasting, with transformer-based methods becoming increasingly prevalent. Since stock return prediction is widely regarded as a time-series forecasting problem, we included a series of transformer-based models to test the most popular methods' effects over this task. The vanilla Transformer (Vaswani et al., 2017) was included as a baseline. Pytorch implementation of LogSparse (Li et al., 2019), Reformer (Kitaev et al., 2020), and Informer (Zhou et al., 2021), which improve structure of the self-attention mechanism from vanilla transformer to enhance effciency and accruacy, were adopted for comparison. These methods apply sparse version of self-attention mechanism to improve the ability to discover long-range dependencies. Additionally, we explored innovative approaches such as Autoformer (Wu et al., 2021), which incorporates traditional time-series seasonal-trend decomposition techniques and replaced self-attention with autocorrelation to analyze sequence lags. Fedformer (Zhou et al., 2022), which adopts Autoformer's

decomposition strategy and combines Fourier analysis, was also integrated. Finally, Crossformer (Zhang & Yan, 2023) was incorporated for its novel cross-dimensional feature attention, illuminating previously unexplored interdependencies.

**IV. GNN-Based Methods**    Graph neural network (GNN) offers a novel approach to stock market analysis by incorporating information from other stocks into individual stocks based on relationships between them. These methods typically construct a market graph, identifying related stocks as neighbors and leveraging their combined features for enhanced stock return predictions. We adopted the Graph Attention Networks (GAT) (Veličković et al., 2018) as the foundational model for this type of approach. We also included recent graph-based methods: STHAN-SR (Sawhney et al., 2021) and ALSP-TF (Wang et al., 2022). STHAN-SR uses text data and industry classifications to map the stock market graph, implementing an LSTM + Attention model for prediction. Meanwhile, ALSP-TF introduces a data-driven approach, using Dynamic Time Warping (DTW) (Jeong et al., 2011) to establish stock relationships based solely on their features, overcoming data availability limitations in traditional methods. CI-STHPAN (Xia et al., 2024) combines both STHAN-SR and ALSP-TF's approaches of constructing graph and adds patching technique in attention mechanism. MASTER (Li et al., 2024) further explores the data-driven approach by using an attention mechanism to form a graph and incorporate market information for prediction.

**IV. Other Methods**    Alongside the four major types of methods mentioned above, we included other methods that do not fit into these categories but offer intriguing approaches to stock prediction. We included two variations of linear method (Zeng et al., 2023) that challenged the effectiveness of transformers. The study used two simple linear methods DLinear and NLinear and tests over time-series forecast datasets, which achieve better outcome compared to previous transformer-based methods. This led to the conclusion that transformers are not effective for time series forecasting. As a result, we included these two methods to check the validity of such claim under stock return prediction task. FactorVAE (Duan et al., 2022) applies VAE to identify effective latent factors from highly noised features extracted by GRU (Chung et al., 2014) network, adding a new approach in learn effective factors in stock data. Mamba (Gu & Dao, 2024) is the latest method that uses structured state space models (SSMs) to improve Transformers' computational efficiency. It avoids using attention mechanism but manages to perform even better than Transformer-based state-of-art methods in multiple tasks. We included Mamba for testing the effects of latest methods on stock data.

The detail of datasets and metrics used in each method is summarized in Table 7.

| Method | Venue | Dataset | Market | Data Source | Test period | Frequency | Metrics |
|---|---|---|---|---|---|---|---|
| LR | Rev. Financ. Stud. 2020 | NYSE/NASDAQ/AMEX | US | CRSP | 1987-2016 | Month | AR/SR/$R^2$ |
| GBRT | Rev. Financ. Stud. 2020 | NYSE/NASDAQ/AMEX | US | CRSP | 1987-2016 | Month | AR/SR/$R^2$ |
| RF | Rev. Financ. Stud. 2020 | NYSE/NASDAQ/AMEX | US | CRSP | 1987-2016 | Month | AR/SR/$R^2$ |
| MLP | Rev. Financ. Stud. 2020 | NYSE/NASDAQ/AMEX | US | CRSP | 1987-2016 | Month | AR/SR/$R^2$ |
| LSTM | Neural Comput. 1997 | NASDAQ 100 | US | UCSD | 2016 | Minute | MAE/MAPE/RMSE |
| DA-RNN | IJCAI 2017 | NASDAQ 100 | US | UCSD | 2016 | Minute | MAE/MAPE/RMSE |
| AdaRNN | CIKM 2021 | Private stock dataset | Unknown | Unknown | 2017-2019 | Unknown | IC/ICIR/RankIC/RankICIR |
| Transformer | NeurIPS 2017 | Time-series benchmark | - | - | - | - | MSE/MAE |
| LogSparse | NeurIPS 2019 | Time-series benchmark | - | - | - | - | MSE/MAE |
| Reformer | ICLR 2020 | Time-series benchmark | - | - | - | - | MSE/MAE |
| Informer | AAAI 2021 | Time-series benchmark | - | - | - | - | MSE/MAE |
| Autoformer | NeurIPS 2021 | Time-series benchmark | - | - | - | - | MSE/MAE |
| Fedformer | ICML 2022 | Time-series benchmark | - | - | - | - | MSE/MAE |
| Crossformer | ICLR 2023 | Time-series benchmark | - | - | - | - | MSE/MAE |
| GAT | ICLR 2018 | NYSE/NASDAQ/TOPIX 100 | US/Japan | Google | 2017/2020 | Day | AR/SR/MDD |
| STHAN-SR | AAAI 2021 | NYSE/NASDAQ/TOPIX 100[4] | US/Japan | Google | 2017/2020 | Day | AR/SR/MDD |
| ALSP-TF | IJCAI 2022 | NYSE/NASDAQ/TOPIX 100[4] | US/Japan | Google | 2017/2020 | Day | AR/SR/MDD |
| CI-STHPAN | AAAI 2024 | NYSE/NASDAQ | US | Google | 2017 | Day | AR/SR |
| MASTER | AAAI 2024 | CSI 300/CSI 800 | China | Yahoo | 2020-2022 | Day | IC/ICIR/AR/IR |
| NLinear | AAAI 2023 | Time-series benchmark | - | - | - | - | MSE/MAE |
| DLinear | AAAI 2023 | Time-series benchmark | - | - | - | - | MSE/MAE |
| FactorVAE | AAAI 2022 | A-share | China | Yahoo | 2019-2020 | Day | Rank IC/ Rank ICIR |
| MAMBA | arxiv 2024 | DNA/Audio datset | - | - | - | - | Accuracy (%) |

Table 7: Summary of the methods included in *BenchStock*

A.5 EVALUATION METRICS

We evaluated stock prediction results using a backtesting program from Microsoft's Qlib, which is designed to simulate real-world trading scenarios. Unlike traditional error-based metrics like MSE, Qlib assesses forecasts by forming portfolios and applying return-related financial metrics to gauge performance considering the purpose of model prediction is for portfolio management. The following finance metrics are used for measuring portfolio performance:

- Annual Return (AR): This is the geometric average of the annual returns realized from the portfolio over the evaluation period. The equation is shown as below:

$$\text{Annual Return} = \left( \prod_{i=1}^{n} (1 + R_i) \right)^{\frac{252}{n}} - 1 \tag{2}$$

  where $n$ is the number of trading days. 252 is an estimation of trading days in 1 year.

- Sharpe Ratio (SR): This measures the ratio of excess returns to volatility, serving as a standard metric for evaluating the risk-adjusted returns of an investment. A higher Sharpe ratio indicates greater returns per unit of risk. The equation is shown as below:

$$\text{Sharpe Ratio} = \frac{AR - R_f}{\sigma} \tag{3}$$

  where $R_f$ represents risk-free rate and $\sigma$ is the annually standard deviation of the portfolio.

- Information Ratio (IR): This measures portfolio's ability in generating excess returns relative to a benchmark, while also considering the risk taken to achieve those returns. The equation is shown as below:

$$\text{Information Ratio} = \frac{AR - AR_{benchmark}}{\sigma_{portfolio-benchmark}} \tag{4}$$

  where $AR_{benchmark}$ is the annual return of a benchmark, and $\sigma_{portfolio-benchmark}$ is the annualized standard deviation of differences between portfolio returns and benchmark returns. Information ratio is almost the same as sharpe ratio, except that Sharpe ratio compares against risk free rate, while information compares against benchmark.

- Maximum Drawdown (MDD): This represents the largest peak-to-trough decline of portfolio value during the trading period. A smaller in magnitude Maximum Drawdown indicates a more stable investment. In this paper, we display MDD in negative value, so the higher the value means better performance. We calculate the average of annual MDD over the evaluation period.

Note that our calculation of metrics is different from Qlib's built-in function. Qlib simply takes arithmetic mean of returns within annual return, sharpe ratio and information ratio, which deviates from their original definitions in finance and will lead to invalid values like annual return lower than -100%. In our evaluation, we corrected all the metrics above to ensure the validity.

---

[4]The NYSE and NASDAQ datasets in these works are not the complete version. They only include stocks continually traded from 2013 to 2017.

## A.6 EVALUATION METHOD

To ensure a more comprehensive evaluation that closely reflects reality, we tested the methods as follows. First, portfolios were constructed based on daily forecasts and updated using a top-k strategy. We started with an initial capital of 100 million, maintaining 50 stocks and replacing $k = 10$ at the end of each trading day. This approach differs from previous studies, which typically involved buying and selling 5 stocks daily and calculating the average return from those stocks (Sawhney et al., 2021; Wang et al., 2022). Our strategy accounted for trading costs and stability, as frequent trading can lead to high commission fees and increased volatility. Second, Qlib considered stock suspensions based on our dataset and adhered to regulations specific to each market to enhance realism. Third, we extended the evaluation period from the shorter spans common in prior research to three decades, providing insights into both long-term and short-term model performance. It is important to note that stock prediction results can be more random compared to tasks with higher signal-to-noise ratios; thus, short-term performance does not guarantee long-term effectiveness. Therefore, we evaluated both long-term and short-term performance comprehensively. For the U.S. market, we analyzed data from 1994 to 2023, while for the Chinese market, the analysis covered 1996 to 2023, given that the market primarily began in 1991. For consistent datasets in both markets, methods were assessed from 2005 to 2023.

## A.7 HYPERPARAMETERS

The hyperparameters of each method is summarized in Table 8.

| **Model Parameters** | | | | | | |
|---|---|---|---|---|---|---|
| GBRT | max iteration | 50 | max depth | 10 | | |
| RF | ntrees | 10 | max depth | 10 | min split | 10000 |
| | min leaf | 10000 | | | | |
| MLP | hidden dims | 128 64 32 | | | | |
| LSTM | hidden layer | 1 | hidden dim | 32 | | |
| DA-RNN | hidden layer | 1 | hidden dim | 32 | | |
| AdaRNN | hidden dim | 64 64 | bottleneck width | 64 | win len | 0 |
| | trans loss | adv | dw | 0.5 | pre_epoch | 10 |
| Transformer | encoder layer | 1 | decoder layer | 1 | n_heads | 1 |
| | d_model | 512 | d_ff | 2048 | | |
| LogSparse | encoder layer | 1 | decoder layer | 1 | n_heads | 1 |
| | d_model | 512 | d_ff | 2048 | sparse flag | True |
| | qk_ker | 4 | v_conv | 0 | | |
| Reformer | encoder layer | 1 | decoder layer | 1 | n_heads | 1 |
| | d_model | 512 | d_ff | 2048 | | |
| Informer | encoder layer | 1 | decoder layer | 1 | n_heads | 1 |
| | d_model | 32 | d_ff | 32 | | |
| Autoformer | encoder layer | 1 | decoder layer | 1 | n_heads | 1 |
| | d_model | 512 | d_ff | 2048 | label_len | 32 |
| | pred_len | 1 | moving_avg | 25 | | |
| Fedformer | encoder layer | 1 | decoder layer | 1 | n_heads | 1 |
| | d_model | 512 | d_ff | 2048 | version | Fourier |
| | mode_select | random | modes | 64 | | |
| Crossformer | d_model | 32 | win_size | 2 | seg_len | 2 |
| | n_heads | 1 | | | | |
| GAT | hidden dim | 32 | gl | 0 | alpha | 0.1 |
| STHAN-SR | negative_slope | 0.2 | gl | 0 | alpha | 0.1 |
| ALSP-TF | ws | 3 | gl | 0.01 | hidden dim | 32 |
| CI-STHPAN | context_window | 64 | target_window | 1 | n_layers | 1 |
| | n_heads | 1 | d_model | 32 | d_ff | 512 |
| MASTER | d_model | 32 | t_nhead | 4 | s_nhead | 2 |
| | T_dropout_rate | 0.5 | S_dropout_rate | 0.5 | gate_input_start_index | 7 |
| | gate_input_end_index | 13 | beta | 1.0 | | |
| FactorVAE | hidden layer | 1 | hidden dim | 32 | | |
| MAMBA | d_model | 32 | d_state | 2 | d_conv | 64 |
| | expand | 1 | | | | |

Table 8: Summary of the hyperparameters used for each method

## A.8 CUMULATIVE RETURNS

The following graphs are the log(cumulative return) of each method on four datasets.

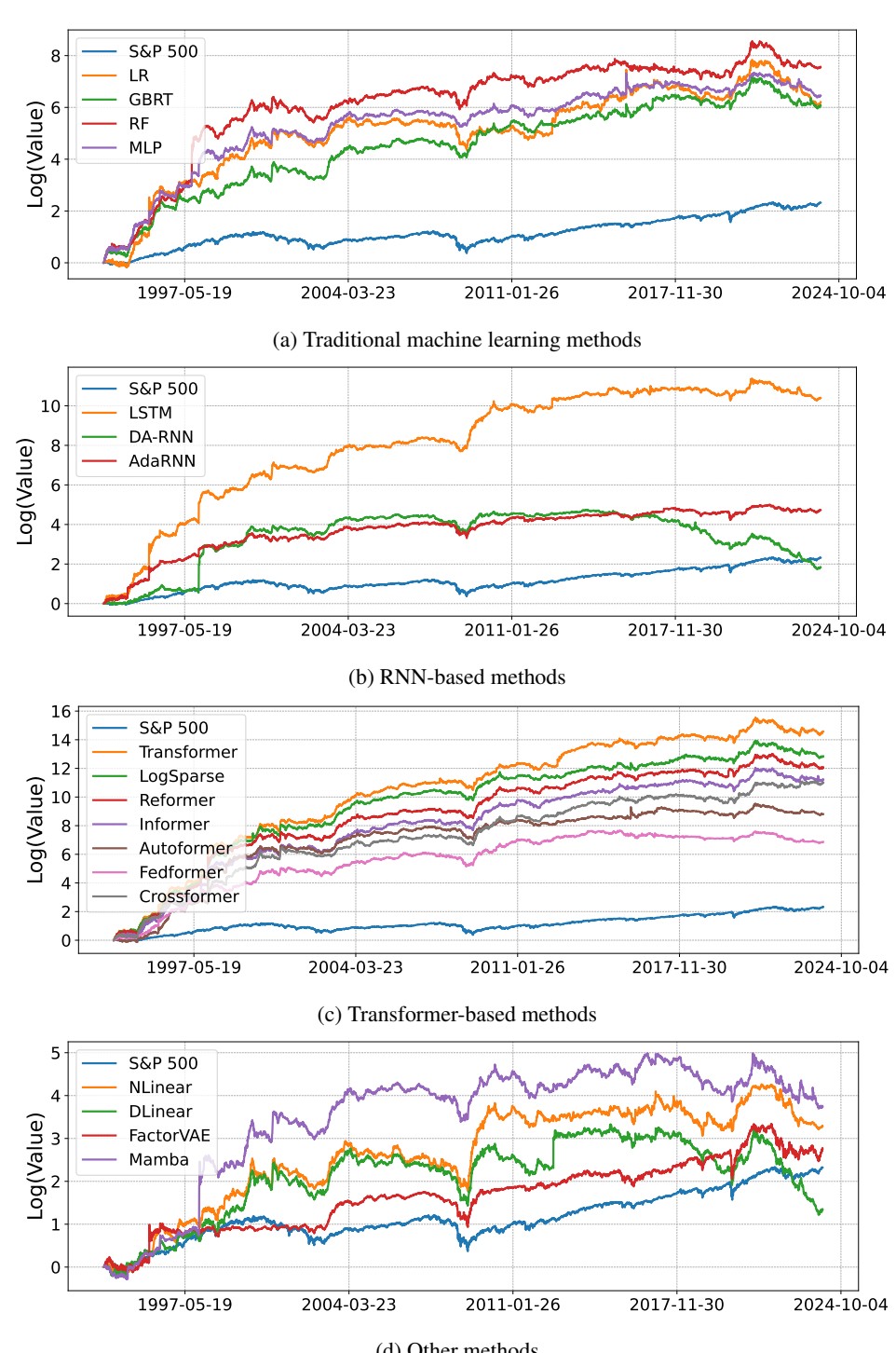

(a) Traditional machine learning methods

(b) RNN-based methods

(c) Transformer-based methods

(d) Other methods

Figure 3: Comparison of log(Cumulative Portfolio value) from 1994 to 2023 on the US-Full dataset. For comparison, we included S&P 500 as the benchmark.

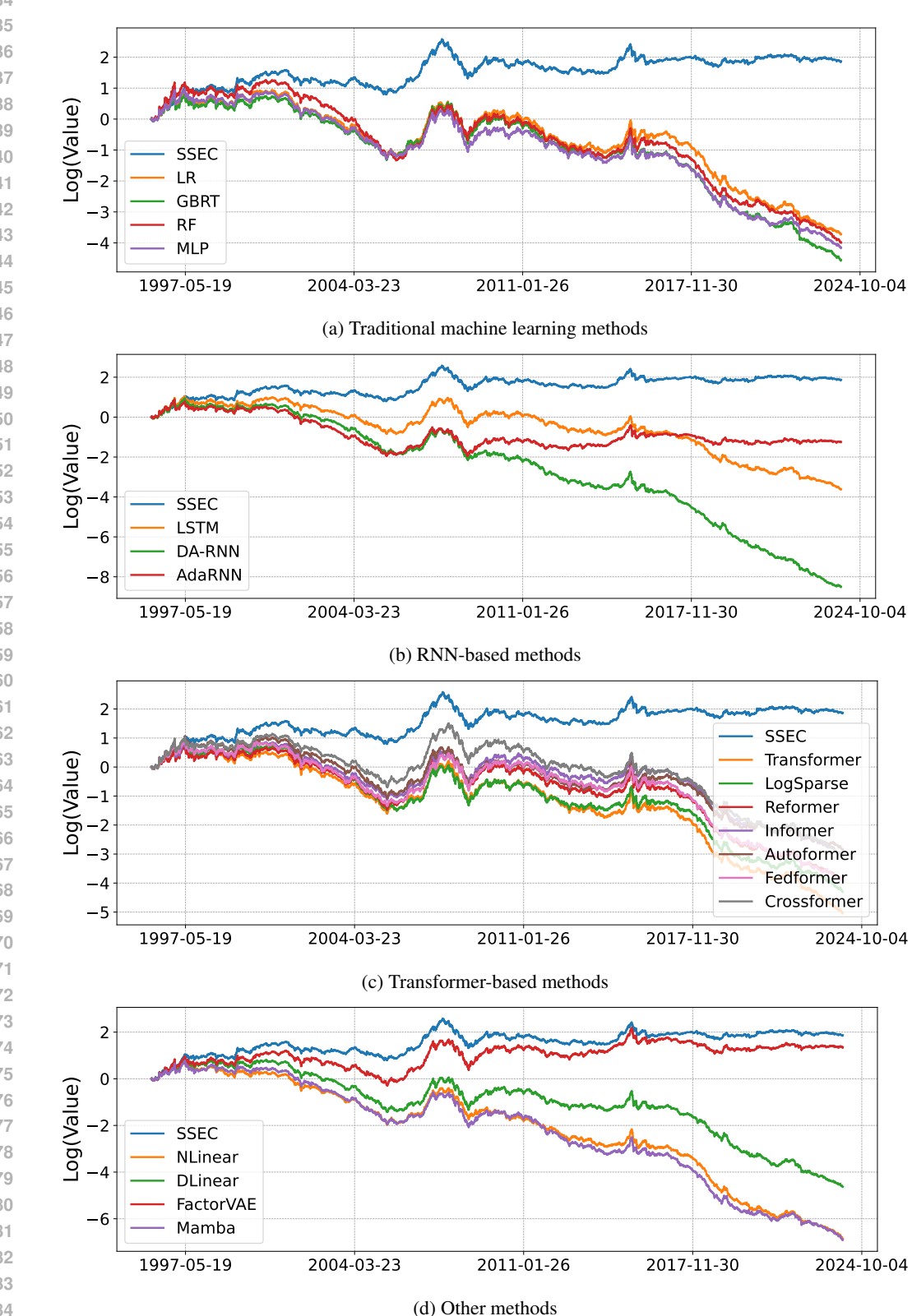

Figure 4: Comparison of log(Cumulative Portfolio value) from 1996 to 2023 on CN-Full dataset. For comparison, we include SSEC as the benchmark.

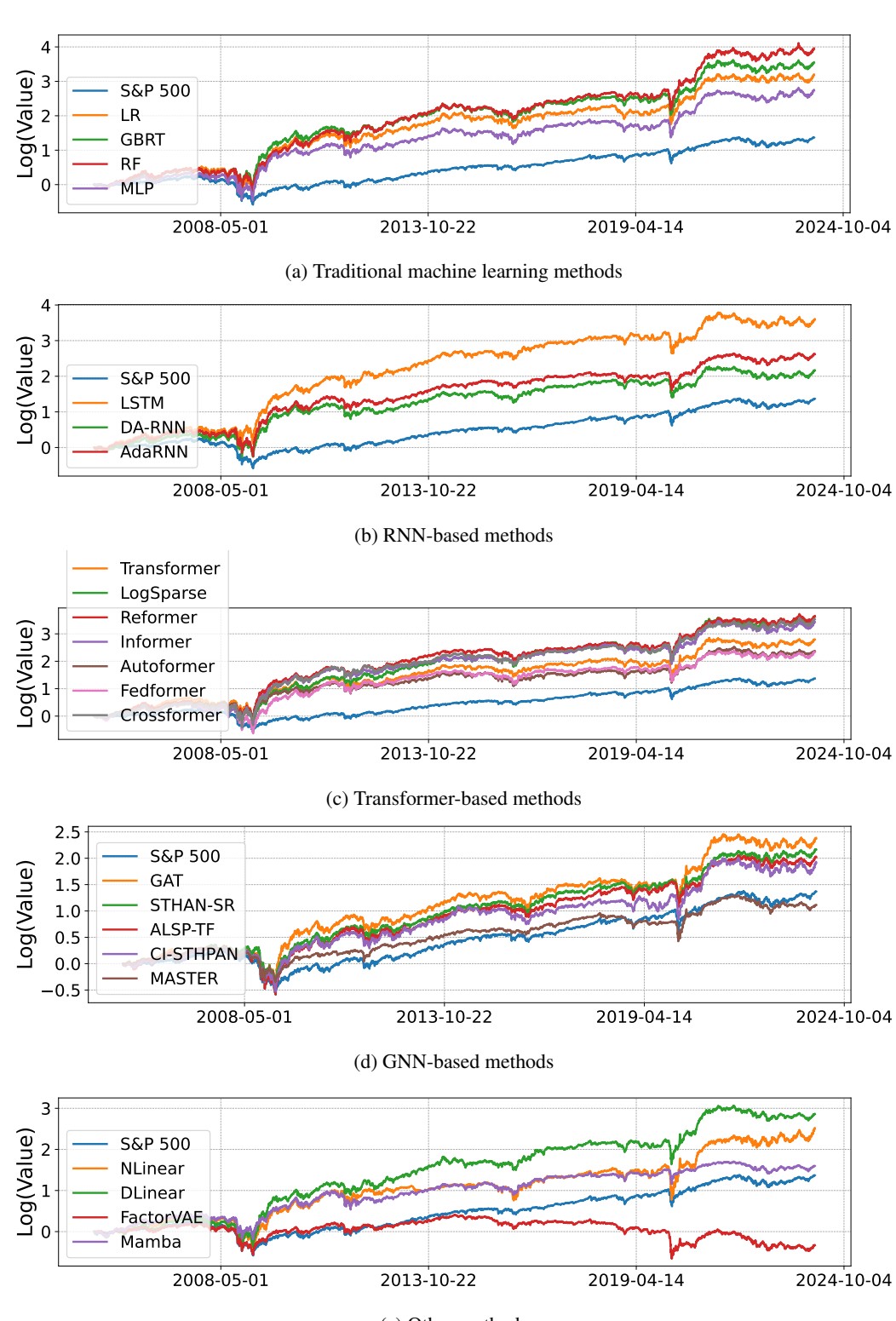

Figure 5: Comparison of log(Cumulative Portfolio value) from 2005 to 2023 on the US-Con dataset.

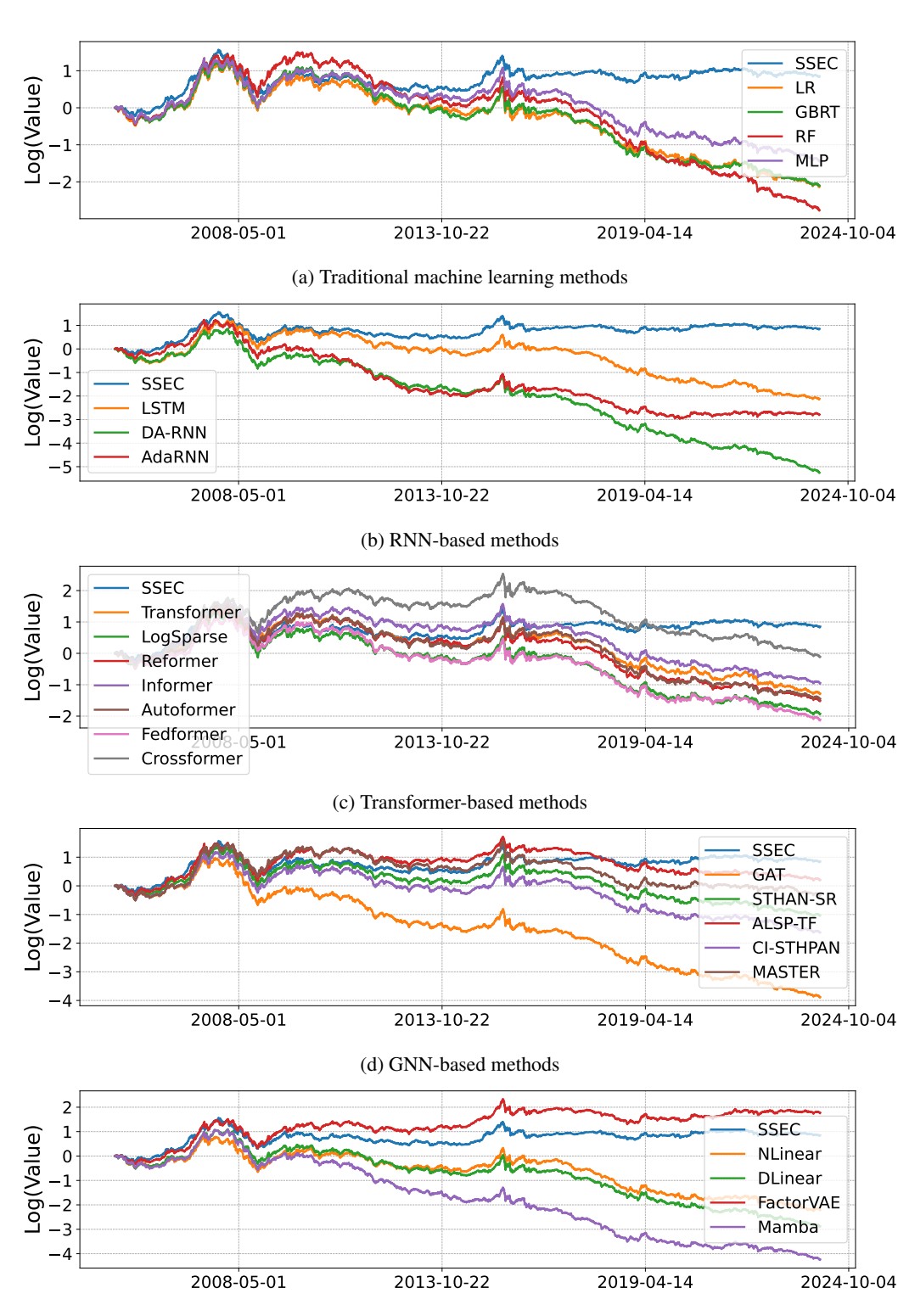

(a) Traditional machine learning methods

(b) RNN-based methods

(c) Transformer-based methods

(d) GNN-based methods

(e) Other methods

Figure 6: Comparison of log(Cumulative Portfolio value) from 2005 to 2023 on CN-Con dataset.

## A.9    Information Coefficient

To investigate the reasons behind the differing performances of various methods across the two markets, we assessed prediction accuracy using the Information Coefficient (IC), which measures the correlation between predicted and actual returns. The IC ranges from -1 to 1, with higher values indicating better accuracy.

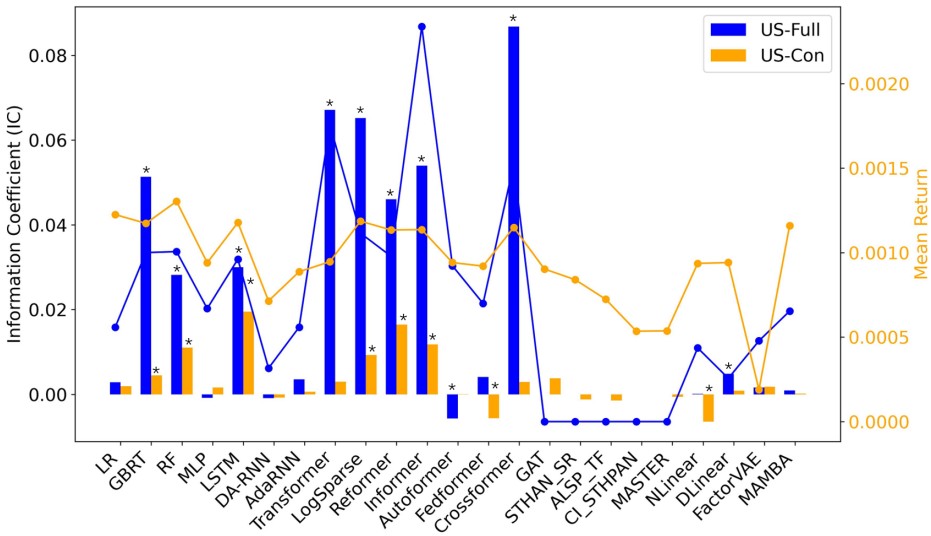

Figure 7: Comparison of IC between full dataset and consistent dataset in the U.S. market.The "*" sign in Figures 7 represents the IC of the method is statistically significant different from 0 at 0.05 level on the corresponding dataset. Note that the GNN-based methods have zero value for IC and mean return on full datasets because they were only tested on consistent datasets.

## A.10    Results after 2008

| Methods | US-Full | | | | CN-Full | | | |
|---|---|---|---|---|---|---|---|---|
| | AR(%) | SR | IR | MDD(%) | AR(%) | SR | IR | MDD(%) |
| LR | 5.71 | 0.06 | -0.03 | -35.98 | -23.24 | -0.92 | -1.20 | -37.60 |
| GBRT | 7.08±5.12 | 0.09±0.09 | -0.03±0.12 | -32.47±1.40 | -26.88±3.27 | -1.08±0.12 | -1.54±0.20 | -38.66±1.56 |
| RF | 7.16±0.62 | 0.18±0.02 | -0.02±0.02 | -28.24±0.25 | -23.85±3.41 | -0.92±0.10 | -1.33±0.18 | -37.83±1.82 |
| MLP | 5.02±4.62 | 0.04±0.09 | -0.10±0.12 | -28.63±0.53 | -24.03±3.70 | -0.95±0.13 | -1.26±0.22 | -38.29±1.46 |
| LSTM | 14.51±2.31 | 0.38±0.07 | 0.22±0.07 | -29.38±1.01 | -24.55±3.36 | -0.95±0.10 | -1.25±0.18 | -38.42±1.96 |
| DA-RNN | -13.75±5.00 | -0.56±0.21 | -0.79±0.25 | -33.09±1.81 | -38.39±4.91 | -1.46±0.17 | -2.05±0.24 | -46.26±2.95 |
| AdaRNN | 5.18±1.36 | 0.15±0.06 | -0.23±0.12 | -19.37±0.22 | -3.59±0.50 | -0.22±0.02 | -0.01±0.04 | -26.28±0.39 |
| Transformer | 22.41±6.09 | 0.40±0.08 | 0.29±0.07 | -32.69±0.92 | -27.67±2.47 | -1.03±0.09 | -1.42±0.14 | -40.37±1.06 |
| Logsparse | 17.24±4.66 | 0.35±0.14 | 0.22±0.11 | -32.80±0.93 | -23.64±1.25 | -0.89±0.04 | -1.20±0.07 | -38.20±0.68 |
| Reformer | 17.35±8.63 | 0.44±0.24 | 0.29±0.26 | -31.54±1.54 | -24.06±1.85 | -0.92±0.06 | -1.29±0.10 | -38.01±1.18 |
| Informer | 24.13±2.93 | 0.62±0.04 | 0.48±0.04 | -29.95±1.24 | -19.92±1.36 | -0.79±0.05 | -1.04±0.08 | -36.30±1.22 |
| Autoformer | 10.92±7.08 | 0.15±0.13 | 0.01±0.18 | -27.74±3.56 | -19.49±1.52 | -0.80±0.06 | -1.03±0.11 | -35.84±0.31 |
| Fedformer | 5.11±2.61 | 0.13±0.11 | -0.09±0.09 | -25.67±2.88 | -23.79±0.75 | -0.91±0.04 | -1.23±0.06 | -38.48±0.48 |
| Crossformer | 26.30±6.19 | 0.70±0.26 | 0.54±0.22 | -28.48±2.80 | -24.58±3.54 | -0.98±0.12 | -1.31±0.20 | -38.17±2.11 |
| NLinear | 3.46±5.05 | 0.04±0.15 | -0.12±0.15 | -32.78±1.16 | -32.94±3.21 | -1.34±0.15 | -1.81±0.19 | -41.48±1.18 |
| DLinear | -6.29±3.18 | -0.22±0.12 | -0.39±0.16 | -36.22±1.91 | -25.15±0.25 | -0.97±0.02 | -1.28±0.04 | -39.02±0.26 |
| FactorVAE | 6.14±2.50 | 0.15±0.09 | -0.06±0.09 | -22.47±0.55 | -0.55±15.38 | -0.14±0.56 | 0.03±1.05 | -28.37±2.52 |
| Mamba | 0.05±2.64 | -0.07±0.09 | -0.23±0.12 | -34.50±0.56 | -32.37±0.34 | -1.25±0.01 | -1.69±0.02 | -41.89±0.15 |
| Benchmark | 7.64 | 0.28 | - | -16.25 | -3.50 | -0.24 | - | -23.27 |

Table 9: Evaluation results since 2008 financial crisis over two markets. Annual return and maximum drawdown were shown in percentage. The names of the methods can be found in Sec 3.3.

