# OpenReview forum: "Benchmarking Machine Learning Methods for Stock Prediction"
_ICLR.cc/2025/Conference — ICLR 2025 Conference Withdrawn Submission_

### Official Review · Reviewer_9YjF · 2024-10-24

**Soundness:** 2
**Presentation:** 3
**Contribution:** 1
**Rating:** 3
**Confidence:** 5

**Summary:**

This study conducts a benchmark review with standardized datasets from the U.S. and Chinese stock markets, along with an evaluation framework to analyze the performance of various machine learning models. Through three decades of large-scale experiments, it assesses short- and long-term predictions, evaluates portfolio performance using backtesting, and identifies key insights like market-specific differences and the limitations of advanced models.

**Strengths:**

As a benchmark review study, the selection of models is comprehensive, and the choice of experimental data is rigorous, ensuring reliability and robustness. The paper is well-structured with clear logic, making it easy for readers to follow and understand the content effectively.

**Weaknesses:**

However, I have some concerns regarding the paper's contribution and originality. In fact, some conclusions in the paper appear somewhat arbitrary, lacking more detailed experiments to substantiate them. As a review study, the experiments are limited in scope and lack sufficient dimensions to provide a more comprehensive analysis. Please see more details in Questions

**Questions:**

1, The selection of metrics in this paper presents significant issues, as only finance-related metrics are used, which are inherently subjective and heavily influenced by specific trading strategies and local exchange policies. Furthermore, from a modeling perspective, the paper lacks time-series-related metrics to properly evaluate model performance. This problematic metric selection makes it difficult for readers to determine which model is the most suitable choice.

2, The paper lacks a proper statistical justification for time-series models, which weakens the reliability of the evaluation. While the IC (Information Coefficient) method measures the correlation between predicted and actual returns, it fails to eliminate the possibility that the model's performance is driven by randomness rather than genuine predictive power.

3, From a data perspective, the study lacks testing across multiple frequency dimensions. Focusing on a single frequency is insufficient for a benchmark study, especially in stock prediction, where the evaluation should cover at least high-frequency (minute-level), mid-frequency (hourly-level), and low-frequency (daily-level) predictions to provide a more comprehensive analysis.

4, Similarly, the paper lacks both single-step and multi-step forecasting evaluations. For a benchmark study, it is essential to assess model performance in both scenarios, as single-step tests focus on immediate predictions, while multi-step tests capture a model’s ability to forecast over extended horizons, offering a more complete evaluation of its predictive power.

5, The study fails to incorporate essential features such as news and events, limiting the relevance of the feature set to only open, high, low, and close prices, along with trade volume and trade amount. These features offer limited useful information for the models, and such comparative experiments are unlikely to provide meaningful insights or valuable references for experts in the field.

6, Stock open-source data is easy to access, yet the experimental procedures are incomplete. Regarding the paper’s contribution, it lacks originality and fails to introduce new insights or innovations to advance the field.

**Details Of Ethics Concerns:**

Nan

---

### Official Review · Reviewer_8wJb · 2024-10-26

**Soundness:** 2
**Presentation:** 2
**Contribution:** 1
**Rating:** 3
**Confidence:** 5

**Summary:**

The paper introduces BenchStock, a benchmark for evaluating machine learning methods in stock prediction, addressing the lack of high-quality datasets and comprehensive evaluation methods in the field. It includes standardized datasets from the U.S. and Chinese stock markets and covers models from traditional machine learning to the latest deep learning approaches. Through large-scale experiments over three decades, the study found that most methods outperformed the S&P 500 in the U.S. but not in China, prediction accuracy did not correlate with portfolio return, advanced deep learning methods did not outperform traditional ones, and model performance was highly dependent on the testing period. These findings highlight the complexity of stock prediction and the need for more in-depth research in this area.

**Strengths:**

1. It provides a thorough benchmarking framework for evaluating a wide range of machine learning methods for stock prediction.
2. The paper introduces standardized datasets from two of the world's largest stock markets, the U.S. and China, which helps in making more accurate comparisons between different prediction methods.
3. By conducting experiments over three decades, the study offers insights into both short-term and long-term performance of the prediction models, which is crucial for understanding their real-world applicability.

**Weaknesses:**

1. The study focuses only on price and volume data, which might not capture all relevant factors affecting stock movements. Excluding other data types like news sentiment, economic indicators, or alternative data sources could limit the models' predictive power.
2. While the paper compares performances across two major markets, the findings may not be generalizable to other markets with different characteristics and regulatory environments.

**Questions:**

1. The conclusions and contributions of this study are limited. This paper repeats many existing studies and fail to demonstrate its unique values over existing massive studies. "Most methods outperformed the S&P 500 in the U.S. market but experienced significant losses in the Chinese market." is meaningless. No trading companies would use the same strategy in these two totally different markets. And this observation fails to give any practical implications. "Prediction accuracy of a method was not correlated with its portfolio return." In practice, we care more about the portfolio return, instead of the prediction accuracy metrics. "Advanced deep learning methods did not outperform traditional approaches." Deep learning methods are more useful when news and other external data sources are involved. In the scope of this study, this observation is not reliable. "The performance of the models was highly dependent on the testing period". This observation also fails to state any practical information.
2. Based on "Figure 1: Overall framework of BenchStock", the proposed BenchStock is based on some existing tools and methods. This would let the readers think that BenchStock is a combination of existing mature tools, which is easy and can be done by any parties.
3. The experiments are simply evaluations for existing methods, which is not enough for a top conference publication.

---

### Official Review · Reviewer_Baj7 · 2024-11-02

**Soundness:** 1
**Presentation:** 3
**Contribution:** 1
**Rating:** 3
**Confidence:** 5

**Summary:**

The paper proposes two main contributions.
1. A standardised dataset for comparing stock price prediction algorithms.
2. A comparison of several deep learning methods for stock price prediction accuracy in portfolio optimisation.

**Strengths:**

The paper is sound in terms of its application of machine learning methodology.
The paper is well-written and clear to follow.
The diagrams and tables add value.
The number of methods compared is also quite comprehensive.

**Weaknesses:**

The largest weakness is that the paper doesn't consider existing literature on the problem.
For instance, the 2024 M6 competition https://www.unic.ac.cy/iff/research/forecasting/m-competitions/m6/ involved 100,000 submissions from 50+ countries for stock price prediction and portfolio optimisation on a standardised dataset.
This is the SOTA benchmark for forecasting.

Unfortunately, as a result, the paper offers nothing that isn't already known or doesn't already exist in a significantly better form.

**Questions:**

Why would you consider using portfolio performance as a metric for measuring forecasting performance when it is well-understood in the literature that forecasting volatility is probably more important than point forecasts of price?
https://arxiv.org/abs/2310.13357

---

### Official Review · Reviewer_1poZ · 2024-11-02

**Soundness:** 1
**Presentation:** 1
**Contribution:** 1
**Rating:** 1
**Confidence:** 4

**Summary:**

This study proposes BenchStock to evaluate the performance of ML models for stock price prediction. It consistently preprocessed nearly 30 years of stock market data from the U.S. and China and tested 23 different ML models.

**Strengths:**

The study built and tested a large number of models.

**Weaknesses:**

- Since stock data is mostly publicly available, the value of the data provided in this study is limited. While consistent preprocessing is valuable, it alone may not be enough to establish a unique benchmark.
- Additionally, most ML research on stock prediction uses a much wider variety of datasets beyond stock prices. Therefore, if this benchmark cannot comprehensively include such diverse datasets, it may lack practical applicability.

**Questions:**

- Why did you include only the U.S. and China? Stock prediction for other advanced and developing countries is also a crucial issue, and challenges in data preprocessing or standardization are often even more pronounced in these markets.
- If the benchmark does not incorporate other types of data, such as financial statements, analyst reports, and news articles, it may not be usable by most stock prediction models. Have you considered this limitation?

---

### Official Review · Reviewer_RhfR · 2024-11-04

**Soundness:** 2
**Presentation:** 2
**Contribution:** 2
**Rating:** 3
**Confidence:** 4

**Summary:**

This paper introduces BenchStock, a benchmark that includes standardized datasets from Chinese and US stock markets to benchmark different machine learning and deep learning methods.

**Strengths:**

1. A detailed dataset is introduced in this work with clear
2. Detailed experimental setups are provided to enhance reliability of the results.

**Weaknesses:**

1. It would be better if more insights can be derived from these empirical observations, and how these insights may guide further research. E.g., what might be the underlying mechanism that results in the inferior performance of DL models compared to traditional ML methods (e.g. feature engineering, labeling, etc.)
2. Deep learning methods on CN markets performed quite bad over the years, which seems to contradict with existing works and industrial practices.
3. It would be great to see some new results incorporating higher-frequency data, where the signal-to-noise ratio in the data might be much higher and easier for DL models to learn these patterns.

**Questions:**

Please see above.

---

### Note · Authors · 2024-11-19

I have read and agree with the venue's withdrawal policy on behalf of myself and my co-authors.